# Mouse Models of Musculocontractural Ehlers-Danlos Syndrome

**DOI:** 10.3390/genes14020436

**Published:** 2023-02-08

**Authors:** Takahiro Yoshizawa, Tomoki Kosho

**Affiliations:** 1Division of Animal Research, Research Center for Advanced Science and Technology, Shinshu University, Matsumoto 390-8621, Japan; 2Department of Medical Genetics, Shinshu University School of Medicine, Matsumoto 390-8621, Japan; 3Center for Medical Genetics, Shinshu University Hospital, Matsumoto 390-8621, Japan; 4Division of Clinical Sequencing, Shinshu University School of Medicine, Matsumoto 390-8621, Japan; 5Division of Instrumental Analysis, Research Center for Advanced Science and Technology, Shinshu University, Matsumoto 390-8621, Japan

**Keywords:** musculocontractural Ehler-Danlos syndome (mcEDS), carbohydrate sulfotransferase 14 (CHST14), dermatan 4-*O*-sulfotransferase-1 (D4ST1), dermatan sulfate epimerase (DSE), dermatan sulfate (DS), model mouse

## Abstract

Musculocontractural Ehlers-Danlos syndrome (mcEDS) is a subtype of EDS caused by mutations in the gene for carbohydrate sulfotransferase 14 (*CHST14*) (mcEDS-*CHST14*) or dermatan sulfate epimerase (*DSE*) (mcEDS-*DSE*). These mutations induce loss of enzymatic activity in D4ST1 or DSE and disrupt dermatan sulfate (DS) biosynthesis. The depletion of DS causes the symptoms of mcEDS, such as multiple congenital malformations (e.g., adducted thumbs, clubfeet, and craniofacial characteristics) and progressive connective tissue fragility-related manifestations (e.g., recurrent dislocations, progressive talipes or spinal deformities, pneumothorax or pneumohemothorax, large subcutaneous hematomas, and/or diverticular perforation). Careful observations of patients and model animals are important to investigate pathophysiological mechanisms and therapies for the disorder. Some independent groups have investigated *Chst14* gene-deleted (*Chst14*^-/-^) and *Dse*^-/-^ mice as models of mcEDS-*CHST14* and mcEDS-*DSE*, respectively. These mouse models exhibit similar phenotypes to patients with mcEDS, such as suppressed growth and skin fragility with deformation of the collagen fibrils. Mouse models of mcEDS-*CHST14* also show thoracic kyphosis, hypotonia, and myopathy, which are typical complications of mcEDS. These findings suggest that the mouse models can be useful for research uncovering the pathophysiology of mcEDS and developing etiology-based therapy. In this review, we organize and compare the data of patients and model mice.

## 1. Introduction

Ehlers-Danlos syndromes (EDSs) are a heterogeneous group of heritable connective tissue diseases characterized by joint hypermobility, skin hyperextensibility, and tissue fragility [1,2]. In accordance with the 2017 international classification and subsequent findings, EDSs are classified into 14 subtypes (classical EDS, cardia–valvular EDS, vascular EDS, arthrochalasia EDS, myopathic EDS, classical-like EDS, classical-like EDS type 2, dermatosparaxis EDS, kyphoscoliotic EDS, brittle cornea syndrome, spondylodysplastic EDS, musculocontractural EDS, periodontal EDS, and hypermobile EDS) based on causative genes and/or clinical manifestations [1,3]. In most of these subtypes, mutations have been identified in fibrillar collagen-encoding genes, genes encoding collagen-modifying enzymes, or enzymes that modify glycosaminoglycan chains of proteoglycans [1,3]. Musculocontractural EDS (mcEDS) is a rare subtype of EDS caused by mutations in the genes encoding carbohydrate sulfotransferase 14 (mcEDS-*CHST14*; MIM#601776) or dermatan sulfate epimerase (mcEDS-*DSE*; MIM#615539) (Figure 1) [1,3,4,5,6,7,8]. mcEDS was identified by several independent groups from 2009 to 2010 [9,10,11,12]. A total of 67 patients with mcEDS-*CHST14* and 14 patients with mcEDS-*DSE* had been reported by 2022 [13,14,15]. Both patients with mcEDS-*CHST14* and those with mcEDS-*DSE* show progressive connective tissue fragility-related manifestations, recurrent dislocations, progressive talipes or spinal deformities, pneumothorax or pneumohemothorax, large subcutaneous hematomas, and/or diverticular perforation, while joint manifestations (recurrent/chronic joint dislocations and joint hypermobility), skin features (hyperextensibility, bruisability, fragility, and atrophic scars), constipation, hypotonia, and motor developmental delay were significantly less common in mcEDS-*DSE* than in mcEDS-*CHST14* [13,14].

Dermatan 4-*O*-sulfotransferase-1 (D4ST1) and DSE are essential enzymes for the biosynthesis of dermatan sulfate (DS). D4ST1 and DSE are encoded by *CHST14* and *DSE* genes, respectively [16,17]. Chondroitin sulfate (CS) and DS are classified as glycosaminoglycans (GAGs), which are linear polysaccharide chains consisting of repeating disaccharide units. The biosynthesis of DS and CS is initiated by the synthesis of a tetrasaccharide linker region, glucuronic acid-β1-3galactose-β1-3galactose-β1-4xyloseβ1-*O*-(GlcUA-Gal-Gal-Xyl-), on serine residues of specific core proteins of proteoglycans [18,19,20,21]. CS contains disaccharide units consisting of GlcUA and *N*-acetylgalactosamine (GalNAc), which are commonly sulfated at C-4 and/or C-6 of the hydroxy group on GalNAc residues [22,23,24]. DS differs from CS by containing iduronic acid (IdoUA) in place of GlcUA [22,23,24]. DSE catalyzes the conversion from GlcUA to IdoUA in CS/DS [17,22]. D4ST1 catalyzes the transfer of a sulfate group from the sulfate donor 3′-phosphoadenosine 5′-phosphosulfate to the C-4 position of the GalNAc residue in DS chains in vivo and in vitro [16,25,26] (Figure 1). DS and CS chains frequently exist as CS/DS hybrid chains in mammalian cells and tissues and covalently attach to a core protein to form a proteoglycan. Decorin and biglycan are representative core proteins of these GAG chains [27,28]. These proteoglycans connect collagen fibrils via GAG chains and influence the structure and functions of the extracellular matrix [27,29]. DS was not detected in the urine or skin fibroblast of patients with mcEDS-*CHST14* [30,31]. DS was absent or significantly decreased in skin fibroblasts and urine derived from patients with mcEDS-*DSE* [7,13,32]. Thus, it is thought that the loss or a decrease in DS is a critical cause of mcEDS symptoms.

Animal models are essential for investigating human disease mechanisms and developing therapies. *Chst14* and *Dse* gene-deleted mice are expected to be model animals for mcEDS-*CHST14* and mcEDS-*DSE*, respectively. In this review, we organize the published data regarding mcEDS and model animals.

**Figure 1 genes-14-00436-f001:**
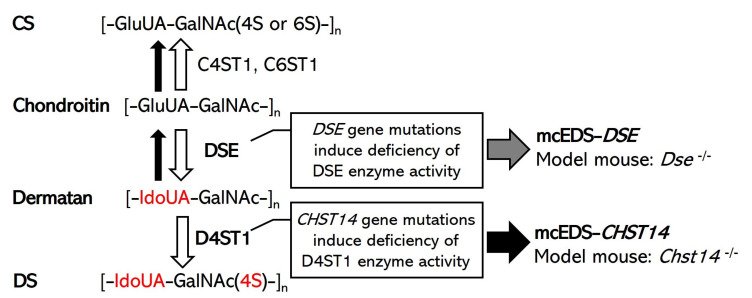
Schematic diagram of the biochemical mechanisms of DS biosynthesis, the pathogenesis of mcEDS, and the expected model mouse. White arrows indicate normal synthesis of DS and CS in healthy (or unaffected) individuals. The gray arrow indicates the consequences of *DSE* gene mutations in patients with mcEDS-*DSE*. Black arrows indicate the consequences of *CHST14* gene mutations in patients with mcEDS-*CHST14*. A defect in D4ST1 enables a back-epimerization reaction that converts IdoUA back to GlcUA to form chondroitin by DSE [17,22].

## 2. Model Mouse of mcEDS-*DSE*

DSE and DSE-like (DSEL) are enzymes responsible for IdoUA formation in the DS chain [17,22,33]. These epimerases are encoded in mice by *Dse* and *Dsel* genes, respectively [34].

### 2.1. Dse Gene-Deleted Mouse

A *Dse* gene-deleted mouse was generated by Maccarana et al. in 2009 [34]. A structural analysis of the CS/DS chains revealed a reduction, but not dissipation, of IdoUA blocks, which are characteristic of DS in the whole body and the skin of *Dse* homozygous gene-deleted mice (*Dse*^-/-^) [34]. Although the *Dse* gene was deleted and DSE expression was not detected in *Dse*^-/-^, partial epimerase activity was observed in some tissues, such as skin, spleen, lung, kidney, and brain tissues, which suggested an influence of enzyme activity of DSEL [34]. The birth rate of *Dse*^-/-^ mixed C57BL/6-129/SvJ genetic background mice was slightly lower than that of Mendelism (offspring of heterozygous breeding pairs: 33% wildtype, 49% heterogeneous gene-deleted, and 18% *Dse*^-/-^) [34]. Conversely, perinatally lethality was observed in *Dse*
^-/-^ mice with a pure C57BL/6 genetic background [35]. Pups of *Dse*^-/-^ mice were smaller (20 to 30% lighter body weight) than their wildtype littermates [34]. Adult *Dse*^-/-^ mice were also smaller (a 5 to 10% shorter crown-to-rump length and 10% lighter) than their wildtype littermates [34]. Kinked tails were observed in both the pups and adults of the mice [34,35]. The embryos of *Dse*^-/-^ mice showed an abdominal wall defect with herniated intestines [35]. The thickness of the skin epidermal layer was increased in newborns of *Dse*^-/-^ pups compared with heterozygote and wildtype littermates [35]. Newborns of *Dse*^-/-^ mice showed an increase in keratin 5 protein expression in the skin basal layer and thicker epidermal layers compared with heterozygote and wildtype littermates [35]. Contents of collagen and decorin, a representative core protein of DS-proteoglycans, were not significantly different from those of the wildtype mice [34]. However, the reduced tensile strength of the skin (41% reduction), with an altered ultrastructure of collagen fibrils, such as irregular outlines and a shift toward thicker fibrils, was observed in *Dse*^-/-^ mice [34]. The immune response against ovalbumin and the migration of dermal dendritic cells to skin-draining lymph nodes were decreased in *Dse*^-/-^ mice [36]. After the ovalbumin injection, there were fewer CD4+ T cells and CD8+ T cells in the spleens of *Dse*^-/-^ mice than in their wildtype littermates [36]. In *Dse*^-/-^ mice, there were also fewer ovalbumin-specific B cells and immunoglobulins than in wildtype littermates [36].

### 2.2. Dsel Gene-Deleted Mouse

Bartolini et al. generated a *Dsel* gene-deleted mouse (*Dsel*^-/-^) in 2012 [37]. The gene expression of *Dsel* is highest in the kidneys and brain, which are tissues with the lowest expression of *Dse* in mice [38]. Epimerase activity and IdoUA content decrease in the brain and kidneys of mutated mice compared with wildtype littermates [37]. However, *Dsel*^-/-^ mice show no anatomical, histological, or morphological abnormalities [37]. The lifespan of the *Dsel*^-/-^ mice was normal [37]. The body weights and lengths of mutated and wildtype littermates were indistinguishable [37]. The brain extracellular matrix architecture did not change in the *Dsel*^-/-^ mice [37].

### 2.3. Double Knockout Mouse of Dse and Dsel

Bartolini et al. (research group of Dr. M. Maccarana) formulated the hypothesis that DSE and DSEL were complementary to each other in mice [34,37]. They generated a double knockout (DKO) mouse with *Dse* and *Dsel* genes (*Dse*^-/-^; *Dsel*^-/-^) by crossbreeding these gene-deleted mice in 2015 [39]. This DKO mouse had complete loss of the IdoUA residue of CS/DS in 2-day-old pups, which suggested a correlation between DSE and DSEL in mice [39]. *Dse* heterogenous knockout (*Dse*^+/-^); *Dsel*^-/-^ mice were mated, and the embryos showed Mendelian ratios on E13.5–E19.5 [39]. Major organs, including the brain, spinal cord, liver, and lungs, in DKO embryos appeared normal [39]. However, the newborns of the DKO mice died in the perinatal period after birth [39]. Some *Dse*^+/-^; *Dsel*^-/-^ and DKO embryos showed umbilical hernia, exencephaly, and a kinked tail [39]. Although the development of secondary lymphoid organs, such as the lymph nodes and the spleen, were normal, the binding of the chemokine (C-X-C motif) ligand 13, an organogenetic modulator of lymphoid, to the DKO embryonic fibroblasts was impaired [39]. The phenotype of connective tissues in DKO mice remains unclear.

## 3. Model Mouse of mcEDS-*CHST14*

In 2010 and 2011, two strains of *Chst14* gene-deleted (*Chst14*^-/-^) mice were generated through homologous recombination targeting the single coding exon of *Chst14* by two independent groups [40,41]. Additionally, Nitahara-Kasahara et al. generated *Chst14* gene-disrupted mice through CRISPR/Cas9-mediated gene editing in 2021 [42]. These *Chst14*^-/-^ mice with a mixed C57BL/6-129/SvJ genetic background had a decreased birth rate [43,44]. Embryonic lethality of *Chst14*^-/-^ mice occurred between E16.5 and E18.5 [44]. Although the percentage of *Chst14*^-/-^ to whole embryos immediately before delivery (E18.5) was 14.8%, only 1.9% of *Chst14*^-/-^ pups survived in childbirth or postnatally [43]. In vitro fertilization and egg transfer did not improve the birth rate of *Chst14*^-/-^ mice [45]. Shimada et al. performed backcrossing of *Chst14*^-/-^ mice to the pure BALB/c genetic background and improved the birth rate of the mice (ratios of offspring of heterozygous breeding pairs before and after backcrossing: 30% from 28% wildtype (*Chst14*^+/+^), 62% from 70% heterogeneous gene-deleted (*Chst14*^+/-^), and 8% from 2% *Chst14*^-/-^) [45]. In an analysis of microsatellite markers, the genetic background of the mouse strain completely replaced the mixed C57BL/6-129/SvJ with BALB/c [45].

Yoshizawa et al. analyzed the prenatal period of the mice [43]. Embryos of *Chst14*^-/-^ (E18.5) mice had a shorter crown-rump length than their *Chst14*^+/+^ and *Chst14*^+/-^ littermates, whereas body weight was unchanged [43]. The gene expression of *Chst14* and the content of DS were significantly decreased in the placentas of *Chst14*^-/-^ embryos [43]. Because the placenta is known to be an organ that is composed of fetal (derived from the epiblast of the *Chst14*^-/-^ embryo) and maternal components (derived from the endometrium of the *Chst14*^+/-^ mother mouse), and thus *Chst14* and DS derived from the maternal placenta were detected, the gene expression of *Chst14* and the content of DS are not completely absent in the analysis of whole placenta. The placentas of *Chst14*^-/-^ embryos exhibited reduced weight and ischemic and/or necrotic-like changes [43]. The percentage volume of the placental villus on the labyrinth zone was significantly decreased in the placentas of *Chst14*^-/-^ embryos [43]. In the placental villus (fetal placenta), an alteration in the vascular structure, with an abnormal structure of the basement membrane of the capillaries, was observed [43]. The basement membranes of capillaries showed decreased thickness and/or rupture [43]. The gene expressions of decorin and biglycan were not different between the whole placentas of *Chst14*^-/-^ and wildtype mice [43]. In immunostaining, biglycan was observed at the capillary basement membranes in both *Chst14*^-/-^ and wildtype mice, whereas decorin was not detected [43]. The gene expression of collagens (*Col1a*, *Col2a*, *Col3a*, *Col4a*, and *Col15*) was unchanged [43]. These findings suggest that *Chst14* is involved in placental development, capillary functions, and fetal survival.

After the perinatal period, the lifespan of viable *Chst14*^-/-^ mice was similar to that of their *Chst14*^+/+^ and *Chst14*^+/-^ littermates [44]. *Chst14*^-/-^ mice had abnormally increased tooth growth, tail deformation (kinked tail), and skin fragility [42,44,46,47]. *Chst14*^-/-^ mice showed a skin fragility-related phenotype, such as a 75% lower level of force required to rupture skin probes than in *Chst14*^+/+^ mice [44,46]. The body and organ weights of the mice were decreased (tibia (41%), heart (37%), liver (34%), and kidney (29%)) compared with *Chst14*^+/+^ mice [44]. Some of these basal phenotypes were similar to all strains of D4ST1-deficient mice generated by the three independent groups.

Hirose et al. performed a detailed histological analysis of the skin from *Chst14*^-/-^ mice [46]. The tensile strength of the skin was decreased to one-fourth in *Chst14*^-/-^ mice compared with *Chst14*^+/+^ mice [46]. The thicknesses of the epidermis and dermis were not different between *Chst14*^-/-^ and *Chst14*^+/+^ [46]. In the dermal reticular layer, the boundary between adjacent collagen bundles was unclear, showing loosened collagen bundles and scattered collagen fibers in *Chst14*^-/-^ mice, whereas the boundary was clear in *Chst14*^+/+^ mice [46]. In transmission electron microscopy, collagen fibrils were oriented in various directions to form disorganized collagen fibers in *Chst14*^-/-^ mice, whereas these fibrils were oriented uniformly to form well-organized fibers in *Chst14*^+/+^ mice [46]. In scanning electron microscopy, disorganized collagen fibrils, which were twisted and turned in various directions, were found in the skin of *Chst14*^-/-^ mice [46]. Through transmission electron microscopy-based cupromeronic blue staining to visualize GAG chains, rod-shaped linear GAG chains were attached, at one end, to collagen fibrils and protruded outside of the fibrils, in contrast to the round and wrapping collagen fibrils in *Chst14*^+/+^ mice [46]. These results suggest that DS deficiency reduces tissue strength with the deformation of GAG chains and collagen fibrils of the extracellular matrix in mice.

Nitahara-Kasahara et al. generated two strains of *Chst14* gene-deleted mice with CRISPR/Cas9-mediated mutations (−1 bp or +6 bp/−10 bp). In these strains, phenotypes similar to other *Chst14*^-/-^ strains, such as growth impairment and skin fragility, were observed [42,44,46]. These mutations in *Chst14* induced the loss of D4ST1 activity and DS deletion [42]. The CS amount was increased sixfold in the skeletal muscles of the mutant mice [42]. The expression of decorin, a core protein of DS, was significantly decreased in the skeletal muscles of *Chst14*^-/-^ mice [47]. Although decorin was present in the muscle perimysium of wildtype mice, decorin was distributed in both the muscle perimysium and endomysium of *Chst14*^-/-^ mice [47]. Mutations in *Chst14* induced myopathy-related phenotypes and thoracic kyphosis [42]. The grip strength of the mutant mice was significantly and consistently weaker than that of heterogeneous mutant and wildtype mice at 2 to 12 months of age [42]. The mutant mice showed significantly decreased motor functions, such as voluntary activity and running speed [42]. In pathological analysis, *Chst14* deficiency induced myopathy phenotypes, including a predominance of small muscle fiber sizes and type I muscle (slow muscle) fibers, compared with *Chst14*^+/+^ and *Chst14*^+/-^ mice [42,47]. Thoracic kyphosis was detected in 1-year-old *Chst14* mutant mice [42]. It was unclear whether the cause of the thoracic kyphosis was an influence of skeletal muscle or a direct effect of DS deficiency in the bone.

Akyuz et al. analyzed the function of DS in the mouse nervous system [44]. The ablation of *Chst14* did not affect brain weight or gross anatomy [44]. In cultured cells of *Chst14*^-/-^ mice, Schwann cells formed longer processes and showed increased proliferation compared with *Chst14*^+/+^ cells [44]. Neurite lengths of cerebellar neurons and motoneurons were significantly increased in the cultured cells of *Chst14*^-/-^ [44]. After femoral nerve injury, functional recovery in *Chst14*^-/-^ mice was significantly accelerated [44]. In *Chst14*^-/-^ mice, the axonal growth rate of motoneurons was accelerated after the injury, suggesting an enhanced regeneration rate [44]. After the injury, the gene expression of CS/DS proteoglycans, such as decorin, biglycan, epiphycan, neuron-glial antigen 2, and tenascin-c, was decreased in the lumbar spinal cord of *Chst14*^-/-^ and *Chst14*^+/+^ compared with that of uninjured mice [44]. In uninjured mice, gene expression of biglycan, tenascin-c, and epiphycan was changed in *Chst14*^-/-^ compared with *Chst14*^+/+^, suggesting an influence of DS deficiency on the composition ratio of core proteins [44]. Rost et al. analyzed the spinal cords of *Chst14*^-/-^ mice [48]. No differences were found in the size of the spinal cords or the numbers of microglia and astrocytes [48]. After severe compression injury of the spinal cord, regeneration was reduced in *Chst14*^-/-^ mice compared with *Chst14*^+/+^ littermates [48]. Li et al. analyzed cognitive functions and hippocampal synaptic plasticity in *Chst14*^-/-^ mice [49]. *Chst14*^-/-^ mice showed deficits in spatial learning and memory, with the reduced expression of hippocampal proteins related to synaptic plasticity [49]. Protein expressions of synaptic proteins in the hippocampus, such as growth-associated protein 43, synaptophysin, N-ethylmaleimide-sensitive factor, N-methyl-D-aspartate receptor (NMDAR) 1, NMDAR2A, NMDAR2B, glutamate receptor 1, and postsynaptic density protein 95, was significantly decreased in *Chst14*^-/-^ mice [49]. The phosphorylation of the intracellular signaling molecules Akt, mTOR, and S6 proteins in the hippocampi was significantly decreased in *Chst14*^-/-^ mice [49].

## 4. Phenotypic Similarities and Differences of Patients with mcEDS and the Mouse Models

It is essential to understand the phenotypic similarities and differences between patients with mcEDS and the mouse models for a deep understanding of the pathophysiology. Most patients with mcEDS (>90%) have various symptoms, such as specific craniofacial (large fontanelle with delayed closure, downward slanting palpebral fissures, and hypertelorism), skeletal (characteristic finger morphologies, joint hypermobility, multiple congenital contractures, progressive talipes deformities, and recurrent joint dislocation), cutaneous (hyperextensibility, fine/acrogeria-like/wrinkling palmar creases, and easily bruised) and ocular (refractive errors) features [14]. Large subcutaneous hematomas, constipation, cryptorchidism, hypotonia, and motor developmental delay are also common (>80%) [14]. In particular, some skeletal, cutaneous, and vascular symptoms (e.g., progressive spinal and joint deformity and subcutaneous hematomas) seriously reduce the activities of daily living (ADL) and quality of life (QOL) of patients [4,14].

On the basis of the causative gene of mcEDS, *Chst14* or *Dse* gene-deficient mice are expected to be typical animal models of the disease. However, mutations in the *DSEL* gene may be linked to bipolar disorder in humans [50]. Because an association between *DSEL* and mcEDS has not been reported, *Dsel*^-/-^ mice and *Dse*^-/-^; *Dsel*^-/-^ mice may not reflect the pathological mechanisms of mcEDS; however, they may be a useful model for investigating the function of IdoUA residue in CS/DS in vivo.

Epimerase activity was completely extinguished, not in *Dse*^-/-^ mice, but in *Dse*^-/-^ and *Dsel*^-/-^ DKO mice, and thus, complementation between DSE and DSEL was suggested [34,39]. Conversely, Müller et al. reported that DSEL cannot compensate for the loss of DSE functions in mcEDS-*DSE* [7]. The amounts of DS disaccharides were decreased by 90.7% and 80.0% in the conditioned medium and the cell fraction from the cultured fibroblasts of a patient with mcEDS-*DSE* compared with a healthy control subject, respectively [7]. Similarly, although the complete loss of epimerase activity was not observed, *Dse*^-/-^ mice showed mcEDS-like phenotypes, such as skin fragility and dermal collagen fibril deformity [34]. These reports suggest that DSE deficiency induces a dramatic decrease in DS and the phenotypes of mcEDS-*DSE* and *Dse*^-/-^ because insufficient compensation with DSEL is inadequate to improve the symptoms.

Comparisons between phenotypes of patients with mcEDS-*CHST14* or mcEDS-*DSE* and *Chst14* and *Dse* gene-deficient mice are shown in Table 1. The embryonic lethality of *Chst14*^-/-^ mice was higher than that of *Dse*^-/-^ mice with a mixed C57BL/6-129/SvJ genetic background [34,43,44]. The body weight of the embryo or pups was reduced by 20–30% in *Dse*^-/-^ mice compared with their wildtype littermates, but it was unchanged in *Chst14*^-/-^ mice [34,43]. Body weights of adult *Dse*^-/-^ and *Chst14*^-/-^ mice were significantly decreased compared with wildtype littermates [34,42,44]. In humans, the influence of mutations in *DSE* or *CHST14* on embryonic survival has not been reported. Patients with mcEDS-*CHST14* showed mild prenatal growth impairment, with a mean birth length of −0.5 SD and a median of −0.6 SD and a mean birthweight of −0.6 SD and a median of −0.67 SD [51]. Postnatal growth was also mildly impaired in terms of slenderness, with a mean height of −0.9 SD and a median of −0.6 SD and a mean weight of −1.5 SD and a median of −1.4 SD [51]. Patients with mcEDS-*DSE* had a normal birth weight [13]. In adults, a mildly decreased body weight (−1.2 SD) and normal height were observed [13].

Skin fragility was observed in 90% and 29% of patients with mcEDS-*CHST14* and mcEDS-*DSE*, respectively [14]. Hyperextensibility of the skin was detected in 100% and 57% of patients with mcEDS-*CHST14* and mcEDS-*DSE*, respectively [14]. The tensile strength of the skin was decreased by 41% and 75% in *Dse*^-/-^ and *Chst14*^-/-^ mice compared with wildtypes, respectively [34,46]. In the pathological observation of the skin specimens, dispersed collagen bundles were similarly observed in patients with mcEDS-*CHST14* or mcEDS-*DSE* as well as in *Chst14*^-/-^ mice [11,13,46,52]. Pathological observation of the skin specimens of *Dse*^-/-^ mice did not mention the dispersion of skin collagen fibrils, but showed an increased diameter of collagen fibrils [34].

One of the most serious symptoms of mcEDS was a large subcutaneous hematoma, which 81% and 67% of patients with mcEDS-*CHST14* and mcEDS-*DSE* had experienced, respectively [4,14]. A subcutaneous hematoma-like phenotype has not been reported in mcEDS model mice. The evaluation of hemorrhage requires attention because the hemostatic system of mice is generally stronger than that of humans. The structure and function of blood capillaries were also regarded as causative factors of subcutaneous hematoma. In *Chst14*^-/-^ embryos, an abnormal structure of the capillary basement membrane in the placenta was observed [43]. Further analysis of blood capillaries in model animals and patients is expected to clarify the mechanisms of subcutaneous hematoma.

Skeletal symptoms seriously affected the ADL and QOL of patients with mcEDS. Patients with mcEDS-*CHST14* and mcEDS-*DSE* had characteristic finger morphologies (both at 100%), joint hypermobility (100% and 67%), multiple congenital contractures (98% and 88%), progressive talipes deformities (98% and 100%), talipes equinovarus (clubfeet) (95% and 75%), recurrent joint dislocations (90% and 60%), and other some skeletal symptoms, respectively [14]. A kinked tail, which suggests skeletal deformity, was observed in both *Dse*^-/-^ and *Chst14*^-/-^ mice [34,44]. *Chst14*^-/-^ mutant mice showed thoracic kyphosis, but not finger deformation in middle age (1 year old) [42]. Investigations of the joints and bones of *Chst14*^-/-^ mice are ongoing.

Neuromuscular symptoms were detected in some mcEDS patients. Hypotonia was observed in 86% of patients with mcEDS-*CHST14* and 75% of those with mcEDS-*DSE* [14]. A tethered spinal cord was observed in 39% of patients with mcEDS-*CHST14* [14]. Relationships between the cranial nervous system and mcEDS-*DSE* are unclear, with only a few case studies. *Chst14*^-/-^ mice showed hypotonia-related phenotypes, such as decreased grip strength, voluntary activity, and running speed [42]. Furthermore, *Chst14*^-/-^ mice showed deficits in spatial learning and memory, while brain weight was unchanged [38,49]. No differences were found in the size of the spinal cords or the numbers of microglia or astrocytes in *Chst14*^-/-^ mice [48]. The neuromuscular phenotype of *Dse*^-/-^ mice has not been clarified.

## 5. Consideration of the Pathogenic Processes of mcEDS

Analysis of *Chst14*^-/-^ mice has made steady progress, and models of severe mcEDS symptoms (e.g., skin fragility and skeletal deformity) have been investigated. Only one report has focused on the skin phenotype of *Dse*-deleted mice [34]. Gene mutations in *CHST14* and *DSE* induced DS deficiency in common [7,11]. DS deficiency is a common pathophysiological cause of mcEDS-*CHST14* and mcEDS-*DSE*. Because *DSE* and *CHST14* are both related to DS synthesis, the phenotype of *Chst14*^-/-^ mice may also be useful to understand the mechanisms of mcEDS-*DSE* symptoms.

Patients with mcEDS-*DSE*, mcEDS-*CHST14*, and *Chst14*^-/-^ mice showed various common symptoms, such as growth impairment, skin fragility, spinal and talipes deformities, hypotonia, and motor developmental delay [11,13,14,42,43,44,46,51,52]. Growth impairment and skin fragility were also reported in *Dse*^-/-^ mice [34]. The pathogenic process of skin fragility has been well investigated in patients and mouse models. Patients with mcEDS-*DSE* and mcEDS-*CHST14*, and *Chst14*^-/-^ and *Dse*^-/-^ mice, showed skin hyperextensibility and fragility with collagen fibril deformity [11,13,34,46,52]. The extracellular matrix is involved in tissue strength [53]. Collagen is known as an important component of the extracellular matrix, and mutations in fibrous collagen genes (*Col1a1*, *Col1a2*, *Col3a1*, *Col5a1*, or *Col12a1*) induce tissue fragility [1]. It was suggested that DS deficiency induced skin hyperextensibility and fragility with structural changes of the GAG chains and collagen fibrils in the mice [46]. These reports suggest that DS is an important factor for tissue strength and morphology (Figure 2).

Mechanical changes of the tissues caused by DS deficiency are insufficient to explain all pathogenic processes of the symptoms of the patients and the mice. GAGs act as co-receptors at the cell surface and interact with cell growth factors [54]. DS is an activating factor and/or co-receptor of fibroblast growth factors (FGFs) in vitro [55,56]. The suppression of FGF signaling via mutations in FGF receptor 3 causes achondroplasia, which involves growth impairment [57,58,59,60]. These reports suggest that DS influences the activity of cell growth factors and is related to growth impairment in mcEDS (Figure 2).

Patients with mcEDS and *Chst14*^-/-^ mice showed hypotonia and motor developmental delay [14,42]. Motor functions are provided by skeletal muscles and the nervous system. In the skeletal muscles of *Chst14*^-/-^ mice, Nitahara-Kasahara et al. found muscle growth suppression with the upregulated expression of myostatin, a negative regulator of protein synthesis in the muscle [47]. The skeletal muscle of *Chst14*^-/-^ mice showed fibrosis with increased expressions of transforming growth factor-β (TGF-β) and a decreased expression of decorin [47]. A DS deficiency induced cells to form longer processes and increased the proliferation of cultured Schwann cells from *Chst14*^-/-^ mice, suggesting that DS was a regulatory factor of neural development [44]. Peripheral nervous system functions for motor functions in mice have not been elucidated.

These reports suggest that DS is essential for tissue strength, morphology, cell growth, and motor functions (Figure 2).

## 6. Conclusions

In this review, we discussed the reported phenotypes of *Chst14*^-/-^ and *Dse*^-/-^ mice, which were expected to be model animals of mcEDS, and the relationships with symptoms of mcEDS and pathogenic processes. Careful observations of patients and model animals are important to investigate the pathophysiological mechanisms and therapeutic strategies of rare diseases. There are still some unexplained pathogenic processes and molecular mechanisms of mcEDS and the model mice. A more detailed analysis of the mechanisms of the phenotypes and therapies is expected in the future.

## Figures and Tables

**Figure 2 genes-14-00436-f002:**
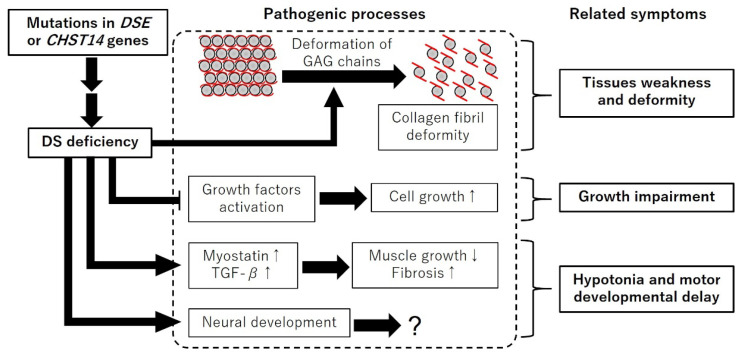
Schematic diagram of the expected pathogenic processes of mcEDS. DS deficiency induces deformation of GAG chains, expression of myostatin and TGF-β, and neural development. DS deficiency inhibits growth factors activation. “↑” and “↓” indicate up and down regulation, respectively.

**Table 1 genes-14-00436-t001:** Comparisons between reported phenotypes of the model mice and symptoms of mcEDS.

	mcEDS-*CHST14*	mcEDS- *DSE*	*Chst14*^-/-^ Mouse	*Dse*^-/-^ Mouse
Perinatal lethality	?	?	+ * [43,44,45]	+ * [34,35]
Reduced body weight	+ [51]	± [13]	− ^#^, + ^$^ [43,44]	+ [34]
Reduced Body length	+ [51]	− [13]	+ [43,44]	+ [34]
Skin fragility	+ (90%) [14]	+ (29%) [14]	+ [42,44,46]	+ [34]
Dermal collagen fibril deformity	+ [11,13,46,52]	+ [11,13,46,52]	+ [46]	+ [34]
Large subcutaneous hematoma	+ (81%) [14]	+ (67%) [14]	− † [43]	?
Spinal deformities	+ (87%) [14]	+ (57%) [14]	+ ^§^ [42,44]	? ^§^ [34,35]
Talipes deformities	+ (98%) [14]	+ (100%) [14]	− [42]	?
Hypotonia	+ (86%) [14]	+ (75%) [14]	+ [42]	?
Motor developmental delay	+ (87%) [14]	+ (75%) [14]	+ [42]	?

The numbers in the parenthesis show the frequency of the symptom among the patients. The square brackets indicate the references. +, positive; −, negative; ?, unclear; *, dependent on genetic background; #, in embryos; $, in adults; †, capillary deformities were observed in the placenta; §, tail deformities were also observed.

## Data Availability

Not applicable.

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
