# Peer review of "Mouse Models of Musculocontractural Ehlers-Danlos Syndrome"

_genes, 2023, doi:10.3390/genes14020436_

Round 1

Reviewer 1 Report

Rare disease is always one of the most important fields of human medical genetics. The manuscript by Takahiro Yoshizawa and Tomoki Kosho is very informative and clear in both style and content. The manuscript has a great potential to be published in GENES. However, some statements need to be clarified to improve the manuscript.

1. L118. Authors stated that "DSE compensates for loss of DSEL in mice". However, An opposite conclusion was made in a paper published in Human molecular genetics, Müller T, Mizumoto S, Suresh I, et al. Loss of dermatan sulfate epimerase (DSE) function results in musculocontractural Ehlers–Danlos syndrome[J]. Human molecular genetics, 2013, 22(18): 3761-3772. In Muller's paper, the conclusion is that "apparently, DSEL cannot compensate for the loss of DSE function", which is apparently different from the author's conclusion in this paper. So, how to reconcile the contradiction? Is there any difference in molecular experiment or method between these studies? Clearly, Muller (2013) paper should be noted and clarified for the different conclusions. 

2. L247 “Investigation of joints and bone of Chst14 -/- mice are ongoing.” Here, the "Investigation" should be "Investigations"

3. It will be very great if authors can visualize the molecular processes through which the two gene defects impact the disease phenotypes. I suggest authors can put a extra figure to show such pathogenic processes in the introduction section. The current figure 1 is good but can be improved by add more details.

4. Overall, the manuscript is near perfect.

Reviewer 2 Report

Yoshizawa and Kosho reviewed recent studies on mouse models of mcEDS-CHST14 and mcEDS-DSE. They compared the data of patients and the mouse models. They concluded that mouse models can be useful in the research for uncovering the pathophysiology of mcEDS and developing etiology-based therapy. Animal models enable scientists to get a clearer picture of the etiology of mcEDS. This review provides an overview of the models. The reviewer, however, have a few concerns:

Major concerns:

1.     The authors introduced that EDSs are classified into 13 subtypes. But there are currently 14 subtypes (see OMIM: Ehlers-Danlos syndrome, classic-like, 2 [MIM: 618000], and PMID: 32732924).

2.     The authors summarized that there are 66 patients with mcEDS-CHST14 and 14 patients with mcEDS- DSE have been reported by 2022. Actually, more patients have been reported, for example, PMID: 36046248 published in 2022. The authors should update the number or clarify why those studies were excluded.

3.     The authors concluded that “Because DSE and CHST14 are both related to DS synthesis, the phenotype of Chst14-/- mice may also be useful to understand the mechanisms of mcEDS-DSE symptoms.” It would be better to further discuss how the authors arrive at this conclusion.

Minor concerns:

1.     The authors should double-check the manuscript to avoid typos. For example, the “Skin flagirity” in Table 1 is supposed to be “Skin fragility”

Round 2

Reviewer 2 Report

The revised manuscript makes contributions to the understanding of mcEDS and is now acceptable.